# Gradient Acceleration in Activation Functions

## Abstract

Dropout has been one of standard approaches to train deep neural networks, and it is known to regularize large models to avoid overfitting. The effect of dropout has been explained by avoiding co-adaptation. In this paper, however, we propose a new explanation of why dropout works and propose a new technique to design better activation functions. First, we show that dropout can be explained as an optimization technique to push the input towards the saturation area of nonlinear activation function by accelerating gradient information flowing even in the saturation area in backpropagation. Based on this explanation, we propose a new technique for activation functions, *gradient acceleration in activation function (GAAF)*, that accelerates gradients to flow even in the saturation area. Then, input to the activation function can climb onto the saturation area which makes the network more robust because the model converges on a flat region. Experiment results support our explanation of dropout and confirm that the proposed GAAF technique improves performances with expected properties.

## 1 Introduction

Deep learning has achieved state-of-the-art performance or sometimes surpassed human-level performance on many machine learning tasks, such as image classification, object recognition, and machine translation (He et al., 2015; Redmon & Farhadi, 2016; Vaswani et al., 2017). To achieve such impressive performance, many techniques have been proposed in different areas: optimization (e.g., Adam (Kingma & Ba, 2014) or Adadelta (Zeiler, 2012)), regularization (e.g., dropout (Hinton et al., 2012)), activation function (e.g., ReLU (Nair & Hinton, 2010)), or layer (e.g., batch-normalization (Ioffe & Szegedy, 2015), Resnet (He et al., 2015)).

To train deep neural networks, regularization is crucial to prevent the models from overfitting and to improve generalization performance. As a regularization method, dropout was proposed to prevent co-adaptation among the hidden nodes of deep feed-forward neural networks by dropping out randomly selected hidden nodes (Hinton et al., 2012; Srivastava et al., 2014). Co-adaptation is a process by which two or more nodes behave as if they are a single node, once the nodes have the same input and output weights. When some nodes are updated and behave together, the model loses some part of its computational power. Dropout is known to break the ties by dropping one of them randomly. Even when dropout is analyzed with probabilistic models, dropout is still assumed to avoid the co-adaptation problem (Baldi & Sadowski, 2013; Kingma et al., 2015; Gal & Ghahramani, 2016). To our best knowledge, the co-adaptation avoidance by dropout has not been thoroughly confirmed yet, even with lots of evidence that dropout improves performance.

On the other hand, it is known that when the outputs of the nonlinear activation function are saturated, the loss function can converge onto a flat region rather than a sharp one with a higher probability (Hochreiter & Schmidhuber, 1997; Chaudhari et al., 2016). Flat regions provide better generalization for test data, because some variation in the input data cannot create a significant difference in the output of layers. However, it is usually hard to train neural networks that have input in the saturation areas of the nonlinear activation functions because there is no gradient information flowing in the areas.

In this paper, we start by questioning the conventional explanation of dropout. Does the effect of dropout come from avoiding co-adaptation only? If not, what are alternative explanations about the effect of dropout? If there are other explanations, can we create another learning techniques based on

such explanations? Basically, our hypothesis is that dropout is an efficient optimization technique rather than a regularizer. We show that dropout makes more gradient information flow even in the saturation areas and it pushes the input towards the saturation area of the activation functions by which models can become more robust after converging on a flat region.

Based on the new explanation, we propose a new technique for the activation function, *gradient acceleration in activation function (GAAF)* that directly adds gradients even in the saturation areas, while it does not change the output values of a given activation function. Thus, GAAF makes models to obtain a better generalization effect. In GAAF, gradients are explicitly added to the areas where dropout generates gradients. In other words, GAAF makes gradients flow through layers in a deterministic way, contrary to dropout which makes gradients stochastically. Thus, GAAF can train the model with less iterations than dropout can.

The paper is organized as follows. Background knowledge including dropout is described in Section 2. In Section 3, we provide a new explanation about how dropout works in terms of optimization. We propose a new technique, GAAF, for activation functions in Section 4. The experiment results are presented and analyzed in Section 5, followed by Section 6 where we conclude.

## 2 BACKGROUND

In this section, we briefly review nonlinear activation functions, dropout, and noise injection in neural networks.

### 2.1 NONLINEAR ACTIVATION FUNCTIONS IN NEURAL NETWORKS

In fully connected neural networks, one layer can be defined as follows:

$$h_j = \phi(z_j), \tag{1}$$

$$z_j = \sum_i W_{ij} x_i + b_j, \tag{2}$$

where $\phi(\cdot)$ is a nonlinear activation function such as sigmoid, tanh, and ReLU. $x_i$ and $h_j$ are input and output for the layer, $W_{ij}$ and $b_j$ are weight and bias, respectively. The sum $z_j$ is referred to as a *net* for the output node $j$. That is, $z_j$ is an input to $\phi(\cdot)$.

In backpropagation, to obtain error information for the node $j$ in the current hidden layer $H_l$, the derivative of $\phi(z_j)$ is multiplied to the weighted sum of the errors $\delta_k$ from the upper layer $H_{l+1}$ as defined in Equation (3).

$$\delta_j = \phi'(z_j) \sum_{k \in H_{l+1}} W_{jk} \delta_k. \tag{3}$$

Note that $\delta_j$ approaches zero when $\phi'(z_j)$ is close to zero, and the amount of gradient information for the weights connected to the node $j$ is proportional to $\delta_j$.

Activation functions have saturation areas where gradients are (almost) zero, which restricts the updating of parameters. When a net value is close to the saturation areas, it is hard to move further since the gradient becomes zero for the net. While the saturation areas hinder training, the functions can play an important role (i.e., nonlinear transformation) around the saturation areas where $\phi(z_j)$ actually provides the nonlinear property.

### 2.2 DROPOUT

Since dropout was proposed in Hinton et al. (2012) to prevent co-adaptation among the hidden nodes of deep feed-forward neural networks, it has been successfully applied to many deep learning models (Dahl et al., 2013; Srivastava et al., 2014). This method randomly omits (or drops out) hidden nodes with probability $p$ (usually $p = 0.5$) during each iteration of the training process, and only the weights that are connected to the surviving nodes are updated by backpropagation. The forward propagation with dropout is defined as follows:

$$z_j = \sum_i W_{ij} d_i x_i + b_j, \tag{4}$$

where $d_i$ is drawn independently from the Bernoulli distribution with probability $p$. When $d_i$ is zero, the input node $x_i$ is dropped out.

After a model with $N$ nodes is trained with dropout, to test new samples, the nodes of the model are rescaled by multiplying $(1 - p)$, which has the effect of taking the geometric mean of $2^N$ dropped-out models. It makes the performance more stable as in the bagging methods (Baldi & Sadowski, 2013). In Hinton et al. (2012); Srivastava et al. (2014); Sainath et al. (2013), it is shown that the neural networks trained with dropout have excellent generalization capabilities and achieve the state-of-the-art performance in several benchmark datasets (Schmidhuber, 2015). In addition to feed-forward layers, dropout can be applied to the convolutional or the recurrent layers. To preserve the spatial or temporal structure while dropping out random nodes, spatial dropout (Tompson et al., 2015) and RnnDrop (Moon et al., 2015) were proposed for the convolutional and the recurrent layers, respectively.

There are a couple of papers that explain how dropout improves the performance (Baldi & Sadowski, 2013; Kingma et al., 2015; Gal & Ghahramani, 2016). Those papers formalize dropout with probabilistic models, assuming that dropout avoids the co-adaptation problem. However, there has been no questioning about co-adaptation avoidance with dropout, to the best of our knowledge.

Interestingly, Ioffe & Szegedy pointed out that batch-normalization could eliminate the need for dropout and they both work towards the same goal as regularizers. If batch-normalization could eliminate the need for dropout, then does batch-normalization reduce co-adaptation? Or was co-adaptation not important for performance improvement by dropout?

In this paper, we question the conventional explanation about the dropout effect, and argue that dropout might be an effective optimization technique, which does not avoid co-adaptation. In addition, dropout usually takes much more time to train neural networks. Thus, if we can replace dropout with a faster technique based on our explanation, the dropout effect can be achieved with a less training time.

## 2.3 Noise Injection to the Network

Like the L1 or L2 norms, regularizers can prevent models from overfitting and improve generalization capability. It has been known that adding noise during training is equivalent to regularizing the model (Bishop, 1995). In addition to dropout, there are several methods to train neural networks with noise, including weight noise injection (Graves et al., 2013), denosing auto-encoder (Vincent et al., 2008), and dropconnect (Wan et al., 2013). Those methods add (or multiply) Gaussian (or Bernoulli) noise to weight (or node) values. For example, weight noise injection adds Gaussian noise to weight values, and dropout multiplies random values drawn from the Bernoulli distribution to node values. Such methods improve performance in many tasks (Graves et al., 2013; Pham et al., 2014).

On the other hand, noise can be applied to activation function as in noisy activation function (Gulcehre et al., 2016). Noisy activation function adds noise where the node output would saturate, so that some gradient information can be propagated even when the outputs are saturated. Although noisy activation function trains the network with noise, it is not explicitly considered as a regularizer. We understand dropout in the same line with noisy activation function, that is, dropout makes gradient flow even in the saturation areas, which is described in the next section.

## 3 Dropout Analysis

In this section, first we show some evidence that the effect of dropout cannot be explained by avoiding co-adaptation. Then, we argue that dropout can work as an effective optimization technique by making more gradient information flow through nonlinear activation functions.

## 3.1 Co-adaptation

To check the presence of the co-adaptation problem, we investigate the correlation between the node values. Generally, correlation between node values is a necessary condition for co-adaptation of the nodes. If dropout avoids co-adaptation, the correlations with dropout should be smaller than the ones without dropout. We checked the correlations between the node values with the MNIST test

dataset after training two feed-forward neural networks with or without dropout. Figure 1 and Table 1 show the distributions of node correlations of each layer and the counts of cases which have high correlation values, respectively.

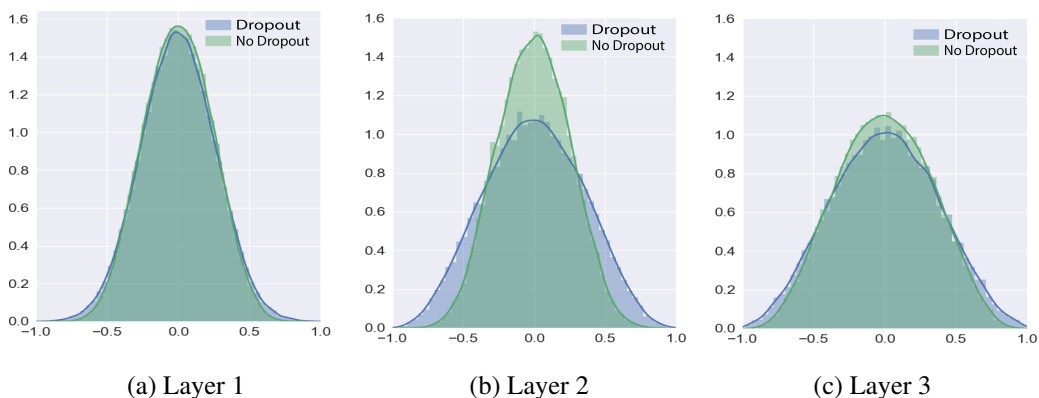

| (a) Layer 1 | (b) Layer 2 | (c) Layer 3 |

Figure 1: Comparison of distributions of node correlations after training two deep network models on MNIST. The horizontal axis indicates the Pearson coefficient and the vertical axis indicate the density.

In Figure 1, the nodes trained with dropout have higher correlations than the nodes trained without dropout, which is against the conventional explanation for the effect of dropout. Actually, dropout increases the degree of correlation between nodes, which indicates high probability of co-adaptation. In Table 1, when the model is trained with dropout, the absolute values of correlation of some node pairs are higher than 0.9, which is not frequently observed in the model trained without dropout. Based on the results, we argue that co-adaptation avoidance is not the best explanation for the dropout effect.

Table 1: The number of cases that the absolute correlation value is higher than a certain value (0.9, 0.8, 0.7, 0.6, and 0.5) out of all the node pairs.

| correlation | Without Dropout | | | | | With Dropout | | | | |
| --- | --- | --- | --- | --- | --- | --- | --- | --- | --- | --- |
| | >0.9 | >0.8 | >0.7 | >0.6 | >0.5 | >0.9 | >0.8 | >0.7 | >0.6 | >0.5 |
| Layer3 | 1 | 123 | 676 | 1989 | 4393 | 174 | 618 | 1625 | 3388 | 6037 |
| Layer2 | 0 | 6 | 71 | 373 | 1305 | 35 | 287 | 1016 | 2541 | 5077 |
| Layer1 | 0 | 5 | 91 | 826 | 3939 | 16 | 181 | 769 | 2510 | 6734 |

## 3.2 OPTIMIZATION

If co-adaptation avoidance is not the best explanation for dropout, then what can explain the performance improvement by dropout? In this section, We argue that dropout generates more gradient information though layers as an optimization technique. The amount of gradient information can be measured by the average of the absolute amount of gradient in each layer. We calculated the gradient information at $k$-th layer $G_k$ with following equation.

$$G_k = \frac{1}{N} \sum_{n=1}^{N} \left( \frac{1}{i * j} \sum_{i,j} \left| \frac{\partial E_n}{\partial W_{i,j}^k} \right| \right), \tag{5}$$

where $N$ is the number of nodes in the layer, and $W_{i,j}^k$ is the weight of the $k$-th layer. To confirm our argument, we compared how much gradient information flows during training models with or without dropout. Table 2 summarizes the amount of gradient information. We can see that dropout increases the amount of gradient information flowing, which is around five times larger than the baseline model.

Then, the next question is how come dropout increases the amount of gradient information. We take a clue from how noisy activation function works (Gulcehre et al., 2016), where the noise allows

Table 2: The amount of gradient information flowing through layers during training. The values in the table are the average value of the absolute value of gradient of all nodes in each layer during the whole iterations.

|        | Without Dropout | With Dropout |
|--------|-----------------|--------------|
| Layer3 | 9.35E-05        | 5.83E-04     |
| Layer2 | 1.40E-04        | 6.52E-04     |
| Layer1 | 1.07E-04        | 5.93E-04     |

gradients to flow easily even when the net is in the saturation areas. Likewise, we believe that dropout could increase the amount of gradient in a similar way.

In deterministic neural networks, the net values are determined with zero variance. However, dropout makes a significant variance for the net, due to the randomness of $d_i$. That is,

$$Var(z_j) = Var(\sum_i^N W_{ij}d_i x_i + b_j) = Var(\sum_i^N W_{ij}d_i x_i) > 0. \tag{6}$$

It also can be empirically confirmed. The node variances from the model for MNIST trained with dropout are summarized in Table 3. To check the variance of net value, $z_j$, we obtained the net values for the same batch 20 times with different random dropout masks during training when the model almost converged. Then, we calculated the variance for the net values of each node and took the average of the variances in each layer. Table 3 presents the average of net variances for one batch (128 data samples) in each layer. Note that the variance of Layer1 is zero, since there is no dropout in the input layer, and Last Layer has a variance generated by dropout in Layer3.

Table 3: The average of net variances in each layer during training with dropout.

|            | Net Variance |
|------------|--------------|
| Last Layer | 1.97         |
| Layer3     | 1.07         |
| Layer2     | 1.07         |

Now, how come the variance helps increase the amount of gradient information? The variance of the net values increases chances to have more gradient information especially around the boundary of saturation areas. In Figure 2, when the derivative $\phi'(z_j)$ is (almost) zero without dropout, there is no gradient flowing through the node. However, if it has a variance, $z_j$ can randomly move to the right or left. In Figure 2, when $z_j$ moves to the right, there is no gradient information as before, but when it moves to the left, it obtains gradient information which is generated by dropout. That is, with a certain amount of probability, gradient information can flow even for $z_j$ in the figure. This phenomenon can explain the dropout effect.

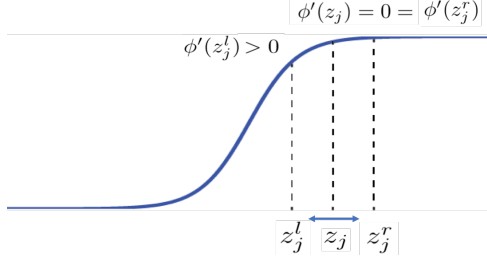

Figure 2: Variance in the saturation area increases the probability to have gradient information, by which dropout generates gradients.

To see whether dropout actually pushes the net values towards the saturation areas, we checked the node value distributions with test data after training. Figure 3 presents the difference between distributions of net values for MNIST test data after training with and without dropout. The model trained with dropout has more net values in the saturation area of $tanh$, which is critical to have

better generalization for test data. Interestingly, the higher layer has more net values in the saturation area, since the variance of the lower layers are transfered to the higher layer.

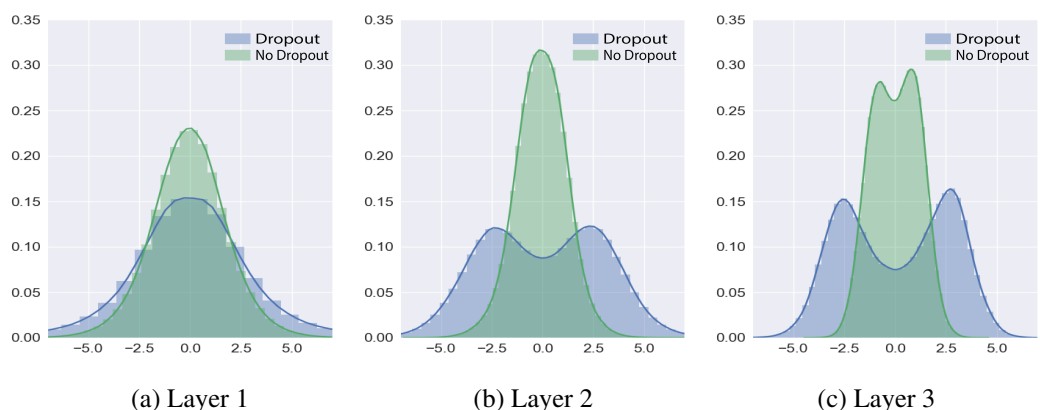

(a) Layer 1          (b) Layer 2          (c) Layer 3

Figure 3: Distributions of net values to $tanh$ for the MNIST test data.

## 4 GRADIENT ACCELERATION IN ACTIVATION FUNCTIONS

Now we understand dropout as an effective optimization technique to increase the amount of gradient information flowing through layers. Then, we are interested in if there is any way to increase gradient information other than dropping out nodes randomly, which takes a lot of time to train the whole networks. We propose a new technique, *gradient acceleration in activation function* (GAAF). The idea is to directly add gradient information for the backpropagation, while not changing (or almost not changing) the output values for the forward-propagation.

Given a nonlinear activation function, $\phi(\cdot)$, we modify it by adding a *gradient acceleration function*, $g(\cdot)$ which is defined by

$$g(x) \quad = \quad (x * K - \lfloor x * K \rfloor - 0.5)/K, \tag{7}$$

where $\lfloor \cdot \rfloor$ is the floor operation, and $K$ is a frequency constant (10,000 in our experiments). Note that the value of $g(x)$ is almost zero ($< \frac{1}{K}$) but the gradient of $g(x)$ is 1 almost everywhere, regardless of the input value $x$. The difference between $\phi(x)$ and the new function $\phi(x) + g(x)$ is less than $\frac{1}{K}$, which is negligible. Figure 4(a) presents what $g(x)$ looks like.

As dropout does not generate gradient information on the leftmost or rightmost saturation areas, we also decrease the gradient acceleration on those areas by multiplying a shape function $s(\cdot)$ to $g(\cdot)$, which leads to our new activation function as follows:

$$\phi_{new}(x) \quad = \quad \phi(x) + g(x) * s(x), \tag{8}$$

where $s(\cdot)$ needs to be defined properly depending on the activation function, $\phi(\cdot)$. For example, when $\phi$ is $tanh$ or $ReLU$, an exponential function or a shifted sigmoid function can work well as $s(\cdot)$, respectively, as shown in Figure 4(b,c). Basically, GAAF can be applied to all kinds of activation functions with a little adjustment of the shape function, which depends on where the saturation areas are located in the activation function.

The proposed gradient acceleration function $g(\cdot)$ generates gradients in a deterministic way, while dropout generates gradient stochastically based on the net variances. Thus, GAAF has the same effect as dropout but it converges faster than dropout. In line with that interpretation, we can understand the different dropout rates Ba & Frey (2013). Generally, if the rate of dropout decreases, then the net variance would decrease, which in turn decreases the amount of gradient on the saturation areas. To obtain the same effect with GAAF, the shape function $s(\cdot)$ needs to be reshaped according to the dropout rate.

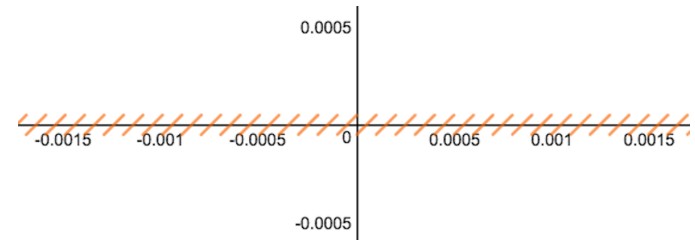

(a) Gradient acceleration function, $g(x)$, which is drawn by the slash lines.

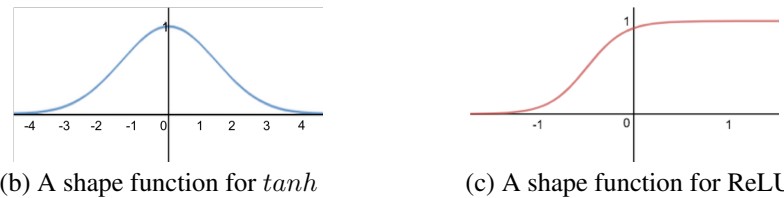

(b) A shape function for $tanh$        (c) A shape function for ReLU

Figure 4: (a) Gradient acceleration function, and (b,c) two shape functions for two activation functions.

## 5 EXPERIMENTS

We evaluate GAAF on several image classification datasets: MNIST (LeCun et al., 1998), CIFAR-10, CIFAR-100 (Krizhevsky & Hinton", 2009), and SVHN (Netzer et al., 2011). The MNIST dataset has hand written digit images (60K train images and 10K test images) of 10 classes (0-9 digits). The CIFAR-10 and CIFAR-100 datasets have 50K train images and 10K test images of 10 and 100 classes, respectively. The SVHN dataset has color images of house numbers from Google Street View and consists of 73K train images and 26K test images of 10 classes (0-9 digits). We use some simple models like DNN for the MNIST and VGG16 (Simonyan & Zisserman, 2014) for the CIFAR and SVHN dataset. In these experiments, we want to check if our GAAF can improve the performance of models, not achieve the state-of-the-art results.

### 5.1 MNIST

To evaluate GAAF on MNIST, we compared three different models: base model, dropout model, and GAAF model. The models have the same architecture, consisting of four feed-forward layers (512-256-256-10) with the $tanh$ activation function. GAAF uses an exponential function as shape function for $tanh$. Table 4 summarizes test accuracies and the number of training epochs for each model to converge. The proposed GAAF model improves the test accuracy as much as the dropout model, while it needs less training epochs than the dropout model.

Table 4: Experiment results on MNIST. The accuracies and epochs are the average values of five executions. The numbers in the parentheses are the corresponding standard deviations.

| Model | Activation | Test Accuracy (%) | Train Epochs |
|---|---|---|---|
| Base Model | tanh | 98.23 (0.075) | 82 (16.6) |
| +Dropout | tanh | 98.40 (0.034) | 169 (9.7) |
| +GAAF | GAAF | 98.35 (0.059) | 114 (24.8) |

In addition, as expected, GAAF increased the gradients flowing through the layers as follows: 1.15E-3, 6.43E-4, 1.78E-4 for the three layers. Compared to the numbers in Table 2, the amounts of gradients by GAAF are as large as by dropout, which are much greater than the amounts by the base model without dropout. Also, we compared the GAAF model and the model without dropout with the the distribution of net values. Figure 5 shows that GAAF pushes the net to the saturation area. Although the difference is not as large as in dropout, but this seems enough to achieve the same level of performance as dropout. Also, by changing the shape function, we expect that we can push the net further, which is our future work.

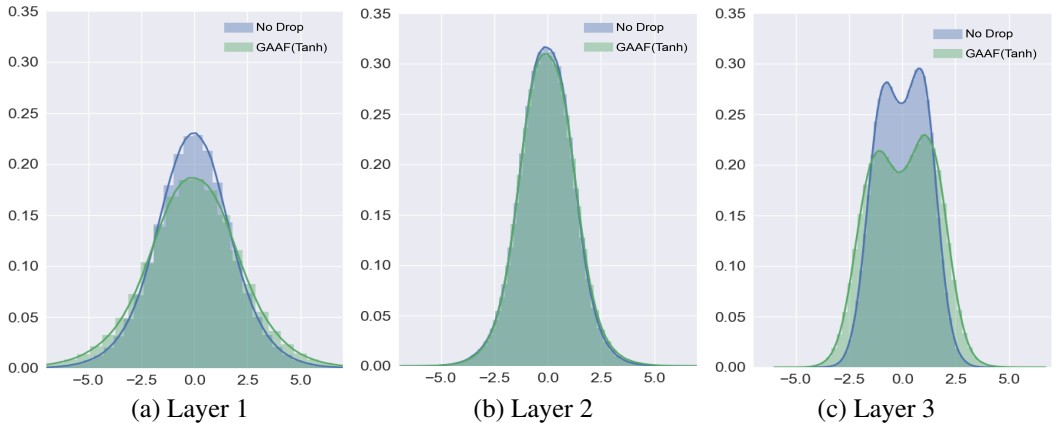

(a) Layer 1          (b) Layer 2          (c) Layer 3

Figure 5: Distributions of net values to $tanh$ for the MNIST test data.

## 5.2 CIFAR AND SVHN

For the CIFAR datasets, we designed a base model similar to VGG16 model (Simonyan & Zisserman, 2014). The model architecture is the same as the original VGG16 model, except that we removed the last three CNN layers and reduced the number of hidden nodes in the feed-forward layers. It is because the CIFAR image size is much smaller than ImageNet.

For the SVHN dataset, we used a simple CNN model as the base model. It has four CNN layers with max pooling after every second CNN layer, and three feed-forward layers on top of the CNN layers.

We evaluated four different models: base model, base model with batch normalization (Ioffe & Szegedy, 2015), GAAF model, and GAAF model with batch normalization. Table 5 summarizes the experiment results on the CIFAR and SVHN datasets. We used ReLU as the activation function for the CNN and feed-forward layers. Thus, GAAF uses a shifted sigmoid function as shape function for ReLU.

Table 5: Test accuracies (%) on CIFAR and SVHN. The numbers are Top-1 accuracies. The improvements achieved by GAAF are presented in the parentheses.

| Model | Activation | CIFAR100 | CIFAR10 | SVHN |
|---|---|---|---|---|
| Base Model | ReLU | 59.63 | 89.55 | 92.03 |
| +Batch Norm (BN) | ReLU | 67.48 | 91.1 | 93.80 |
| +GAAF | GAAF | 61.29 (+1.66) | 90.16 (+0.61) | 92.19 (+0.16) |
| +BN +GAAF | GAAF | **69.36 (+1.88)** | **91.92 (+0.82)** | **94.16 (+0.36)** |

The results confirm that our proposed GAAF improves the base model's performance. More interestingly, GAAF improves performance even with batch normalization, contrary to dropout whose need is eliminated by batch normalization. This shows that GAAF works independently of batch normalization (maybe other optimization techniques too), while dropout hinders batch normalization (or other optimization techniques) by dropping out some (usually the half) of nodes.

In addition, after training, the base and GAAF models have almost the same training accuracies (98.2%, 99.6%, and 99.9% for CIFAR100, CIFAR10 and SVHN, respectively), while GAAF has better test accuracies as shown in Table 5. This supports that GAAF converges on a flat region by pushing the nets towards the saturation areas.

## 6 CONCLUSION

Dropout has been known to regularize large models to avoid overfitting, which was explained by avoiding co-adaptation. In this paper, we presented that dropout works as an effective optimization technique to generate more gradient information flowing through the layers so that it pushes the

nets towards the saturation areas of nonlinear activation functions. This explanation enriches our understanding on how neural networks work.

Based on this explanation, we proposed *gradient acceleration in activation function (GAAF)* that accelerates gradient information in a deterministic way, so that it has a similar effect to the dropout method, but with less iterations. In addition, GAAF works well with batch normalization, while dropout does not. Experiment analysis supports our explanation and experiment results confirm that the proposed technique GAAF improves performances. GAAF can be applied to other nonlinear activation functions with a correspondingly redesigned shape function.

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
