# OpenReview forum: "Gradient Acceleration in Activation Functions"
_ICLR.cc/2019/Conference_

### Official Review · AnonReviewer2 · 2018-11-02
**Interesting idea, but seems less precise than previous work**

**Rating:** 3
**Confidence:** 4

**Review:**

This paper offers the argument that dropout works not due to preventing coadaptation, but because it gives more gradient, especially in the saturated region. However, previous works have already characterized how dropout modifies the activation function, and also the gradient in a more precise way than what is proposed in this paper.

## Co-adaptation
co-adaptation does not seem to mean correlation among the unit activations. It is not too surprising units need more redundancy with dropout, since a highly useful feature might not always be present, but thus need to be replicated elsewhere.

Section 8 of this paper gives a definition of co-adaptation,
based on if the loss is reduced or increased based on a simultaneous change in units.
https://arxiv.org/abs/1412.4736
And this work, https://arxiv.org/abs/1602.04484, reached a conclusion similar to yours
that for some notion of coadaptation, dropout might increase it.

## Gradient acceleration
It does not seem reasonable to measure "gradient information flow" simply as the norm of the gradient, which is sensitive to scales, and it is not clear if the authors accounted for scaling factor of dropout in Table 2.

The proposed resolution, to add this discontinuous step function in (7) with floor is a very interesting idea backed by good experimental results. However, I think the main effect is in adding noise, since the gradient with respect to this function is not meaningful. The main effect is optimizing with respect to the base function, but adding noise when computing the outputs. Previous work have also looked at how dropout noise modifies the effective activation function (and thus its gradient). This work, http://proceedings.mlr.press/v28/wang13a.html, give a more precise characterization instead of treating the effect as adding a function with constant gradient multiplied by an envelop. In fact, the actual gradient with dropout does involve the envelope by chain rule, but the rest is not actually constant as in GAAF.

---

> ### Author Response · Authors · 2018-11-16
> **Thanks for your review.**
>
> Thanks for your helpful review and introducing other previous works related to ours.
>
> In section 2.3, we introduced the noisy activation function and we analyzed that drop out’s effect is quite similar to noise injection in activation function. We also think that our proposed method also quite similar to noisy activation function and we expected the same effect. The noisy activation function increases gradient flow on the saturation areas with stochastic noise, while GAFF does with deterministic method.
>
> For the scale issue that you mentioned, we think it does not affect because we used the same model size and configurations for the Dropout model and the base model.
>
> Also, we think that our analysis of Dropout looks similar to previous works that you introduced, but our contribution is that in line with such analysis, we designed a new activation function which can replace the Dropout layer.

---

### Official Review · AnonReviewer1 · 2018-11-02
**No proper grounding of the presented argument against " avoiding co-adaptation through dropout" concept. Very weak experiments.**

**Rating:** 2
**Confidence:** 5

**Review:**

The authors attempt to propose an alternative explanation for the effect of dropout in a neural network and then present a technique to improve existing activation functions.

Section 3.1 presents a experimental proof of higher co-adaptation in presence of dropout, in my opinion this is an incorrect experiment and request authors to double check. In my experience, using dropout results in sparse representations in the hidden layers which is the effect decreased co-adaptions. Also, a single experiment with MNIST data-set cannot be a proof to reject a theory.

Section 3.2 Table 2 presents a comparison between average gradient flow through layers during training where flow with dropout is higher. This is not very surprising, in my opinion, given the variance of the activation of a neuron in presence of dropout the network tries to optimize the classification cost while trying to reduce the variance. The experimental details are almost nil.

The experiments section 5 presents very weak results. Very little or no improvement and authors randomly introduce BatchNorm into one of the experiment.

---

> ### Author Response · Authors · 2018-11-16
> **Thank you for the feedback.**
>
> Thanks for your helpful review.
>
> We do not want to reject the original explanation of Dropout, but we want to suggest another view of Dropout and within this view, we suggest a new method that can improve the model’s performance.
>
> First of all, we think the sparse representations in hidden layers with Dropout may not be the result of avoiding co-adaptation but the result of better training. A well-optimized model can have sparse representations. We tried to show that the better optimization with Dropout can be explained by increasing gradient information flows in Section 3.
>
> Table 2 proves our explanation about Dropout (the effect of Dropout can be explained by increasing gradient flow rather than avoiding co-adaptation), and the increasing of gradient flow is the basic idea of our proposed method.
>
> In Tabel 5, GAAF improves the model’s accuracy compared to the base model with all three datasets. Also, we tried to show that as we know the effect of traditional Dropout can be eliminated when Batch Normalization (BN) is applied, but GAAF optimizes the model further even with BN. This is why we applied BN to this experiment. The accuracies of BN + GAAF models are higher than BN-only models.

---

> ### Comment · AnonReviewer1 · 2018-11-30
> **No direct reply to concerns.**
>
> Since the authors did not address the concerns in my review directly. I choose to stick to the given rating.

---

### Official Review · AnonReviewer3 · 2018-11-03
**interesting analysis on dropout**

**Rating:** 5
**Confidence:** 3

**Review:**

This paper gives further analysis on dropout and explains why it works although Hinton et al. already showed some analysis. This paper also introduced a new gradient acceleration in activation function (GAAF).

On Table 4, the GAAF is a bit worse than dropout although GAAF converges fast. But i am not sure whether GAAF is really useful on large datasets, not on a small dataset, e.g., MINIST here. On table 5, i am not sure whether you compared with dropout or not. Is your base model already including dropout?

If you want to demonstrate that GAAF is really helpful, i think more experiments and comparisons, especially on larger datsets should be conducted.

---

> ### Author Response · Authors · 2018-11-16
> **Thanks for your review.**
>
> First of all, we thank you for your interests and helpful reviews.
>
> In Table 4, we tried to show that our proposed GAAF can improve the performance of model as much as Dropout does, while the model can be converged much faster than Dropout.
>
> In Table 5, the base model does not include dropout, because we use Convolutional Neural Networks (CNNs). In Tabel 5, we show that the effect of traditional Dropout can be reduced with Batch Normalization (BN), but GAAF works even with BN.
>
> Thank you for your suggestion, we will apply our method to larger datasets like ImageNet. As the effect of Dropout decreases with larger datasets, the effect of GAAF might be reduced with larger datasets. In the current submission, however, our experiments confirm that GAAF helps models optimize better, although it is with small datasets.

---

### Meta-Review · Area_Chair1 · 2018-12-14

**Confidence:** 4
**Recommendation:** Reject

**Metareview:**

Reviewers are in a consensus and recommended to reject. Please take reviewers' comments into consideration to improve your submission should you decide to resubmit.